# Real-Time Depth of Anaesthesia Assessment Based on Hybrid Statistical Features of EEG

**DOI:** 10.3390/s22166099

**Published:** 2022-08-15

**Authors:** Yi Huang, Peng Wen, Bo Song, Yan Li

**Affiliations:** 1School of Engineering, University of Southern Queensland, Toowoomba 4350, Australia; 2School of Mathematics, Physics and Computing, University of Southern Queensland, Toowoomba 4350, Australia

**Keywords:** EEG, depth of anaesthesia, real-time, machine learning

## Abstract

This paper proposed a new depth of anaesthesia (DoA) index for the real-time assessment of DoA using electroencephalography (EEG). In the proposed new DoA index, a wavelet transform threshold was applied to denoise raw EEG signals, and five features were extracted to construct classification models. Then, the Gaussian process regression model was employed for real-time assessment of anaesthesia states. The proposed real-time DoA index was implemented using a sliding window technique and validated using clinical EEG data recorded with the most popular commercial DoA product Bispectral Index monitor (BIS). The results are evaluated using the correlation coefficients and Bland–Altman methods. The outcomes show that the highest and the average correlation coefficients are 0.840 and 0.814, respectively, in the testing dataset. Meanwhile, the scatter plot of Bland–Altman shows that the agreement between BIS and the proposed index is 94.91%. In contrast, the proposed index is free from the electromyography (EMG) effect and surpasses the BIS performance when the signal quality indicator (SQI) is lower than 15, as the proposed index can display high correlation and reliable assessment results compared with clinic observations.

## 1. Introduction

Monitoring the depth of anaesthesia (DoA) during surgery is challenging [1,2]. When anaesthetic agents are applied, the response of the central nervous system could be reflected in the electroencephalography (EEG) [3]. The EEG signals exhibit low-voltage high-frequency and high-voltage low-frequency characteristics in the awake state and deep anaesthesia state, respectively [4]. Thus, the neural activities in the central nervous system could be revealed by EEG. During surgery, the main problem is prescribing precision doses of appropriate anesthetic agents for each patient. An excessively high dosage of anaesthetic agents may lead to coma and postoperative complications, and on the other hand, if the dosage of anaesthetic agents is insufficient, the patient may suffer from pain and waking up [5]. Thus, the real-time DoA assessment can help both patients and anaesthetists.

In the past two decades, the Bispectral Index (BIS) has been widely used for assessing and monitoring the DoA. BIS was developed by Aspect Medical Systems in 1992 [6,7], and is one of the most popular commercial products in the DoA monitoring markets. The BIS index was developed from several complex parameters in the time domain and frequency domain, which are integrated as a one dimension index ranging from 0 (unconscious state) to 100 (awake state) [8,9,10]. Besides, there are some other electrophysiological monitors, such as the Narcotrend index [11], M-Entropy [12], and patient state index [13], for tracing the DoA. However, these DoA monitors still have some limitations, such as inconsistent states between clinical observation and device monitoring index, insensitivity at switch points between consciousness and unconsciousness, and not being accurate across patients [14,15]. Moreover, these real-time monitor systems in clinical use have poor performance when the signal quality index (SQI) is low.

A wide range of features, including frequency domain and time domain features, have been proposed for monitoring DoA over the years. Nguyen-Ky et al. [14] proposed to measure the DoA using a wavelet transform method. In this method, the EEG signal was decomposed into different levels to extract the desired features, and an eigenvector of wavelet coefficients was calculated to develop a new index. Wavelet-weighted median frequency and wavelet coefficient energy entropy methods were used to develop new indexes with high rates of agreement with BIS [16]. Several studies commonly use permutation and sample entropy to create a correlated index with BIS [17,18]. Sarkela et al. [19] used spectral characteristics of the EEG burst suppression as features to propose an automatic method for burst suppression detection and segmentation. Lashkari and Boostani [20] introduced an improved instantaneous frequency (IF) by applying a Kalman filter to develop a noticeable index correlated with the BIS index. In addition, several different types of regression models and classifiers were used to monitor the trending of anaesthetic states, such as an artificial neural network [21,22], an adaptive neuro-fuzzy system [18], a genetic algorithm with a support vector machine (GA-SVM) [23], random forest [24], and convolutional neural network [25]. Neural networks have the advantage of being more flexible in classifying EEG signals. However, these classifiers require a large amount of data to train a robust model, most of the datasets in the research field of anaesthesia are not open to the public, and it is even more challenging to find the public DoA datasets with observation comments and labels from the attending anaesthetists [25].

This study used the threshold value for the wavelet denoising method to pre-process the raw EEG signal data. After feature extraction and selection, five features were evaluated as the input of the Gaussian process regression model to classify different states of anaesthesia. All of the experiments in this study were carried out using MATLAB 2018b and a workstation with an Inter^®^ Core™ i9-10900K CPU @ 3.7 GHz processor.

The first section of the paper provided a brief introduction to the work. Section 2 described the details of the datasets and data acquisition. The pre-processing, features extraction, states of anaesthesia classification, and real-time application are also introduced in this section. Section 3 presents the experiments and results obtained. The discussion on the EMG effect and signal quality were evaluated in Section 4. Section 5 concluded the paper.

## 2. Materials and Methods

The proposed threshold wavelet denoising method started with applying a sliding window technique. The sliding window size was 10 s, having a 9 s overlap with the previous window. Then, the statistical features of the denoised EEG signal were extracted. The regression model was trained by the machine learning method to classify the depth of anaesthesia. Finally, the new index, $N_{DoA}$, was proposed to monitor the DoA in real-time. Figure 1 describes the workflow of the proposed methodology in this study.

### 2.1. Subjects and EEG Recording

The EEG and BIS datasets are directly from our collaborating hospitals, including Toowoomba Hospital, Brisbane Prince Charles Hospital, Queensland, Australia, and our industry partners, such as Shenzhen Delica Medical Equipment Co., Ltd., Shenzhen, China. The EEG and BIS data were obtained using a BIS ${\mathrm{VISTA}}^{\mathrm{TM}}$ monitoring system, version 3.22. All intravenous dosing and intraoperative events were recorded on data log files by the attending anaesthetist. The data log files also contain the BIS index, raw EEG data, signal quality indicator (SQI), impedance, electromyography (EMG), and monitor log of errors. Among the two channels of raw EEG, this study used the one with more consistent EEG quality. In addition, the sampling frequency of the recording data was 128 Hz with a 16-bit signed integer. In this study, a total of 73 subjects with corresponding data were used and the summary of patient demographics is shown in Table 1.

### 2.2. Signal Denoising and Pre-Processing

It is difficult to accurately discriminate the anaesthetic response signal from the raw EEG signal because the noises, such as eye movements, muscle activities, and other artefacts, may corrupt the recorded signals, especially in the awake state. Therefore, all raw EEG signals were pre-processed to reduce the noise. Firstly, the mean and standard-deviation-based threshold methods are used to remove outliers [26]. Then, a wavelet threshold method based on entropy is used to remove the low-amplitude noise and spike noise interference.

Two threshold methods are used to conduct the denoising process, where $T_{h}$ is the threshold value. Donoho [27,28] proposed a threshold wavelet method to remove the noise using a threshold value:(1)$$T_{h}=\sigma \sqrt{2\log N}$$
(2)σ=MAD/0.6745
where σ is the noise standard deviation, *N* is the size of the wavelet coefficient arrays, and *MAD* is the media of absolute detail coefficient of each level in the wavelet transform.

The function of soft-threshold $S_{j}^{\prime }$ has a sign function related to the detail coefficients $C_{j}$.
(3)$$S_{j}^{\prime }=\{\begin{array}{c} sign\left(C_{j}\right)\left(\left|C_{j}\right|-T_{h}\right),\;\;\;\;if\;\left|C_{j}\right|>T_{h}\\ 0,\;\;\;\;\;\;\;\;\;\;\;\;\;\;\;\;\;\;\;\;\;\;\;\;\;\;\;\;\;\;\;\;\;\;\;\;\;\;\;if\;\left|C_{j}\right|\leq T_{h} \end{array}\;$$

The function of hard-threshold $H_{j}^{\prime }$ has a value equal to 0 if the detail coefficient is smaller than the threshold *T*. Otherwise, it is equal to $C_{j}$.
(4)$$H_{j}^{\prime }=\{\begin{array}{c} C_{j},\;\;\;\;\;\;if\;\left|C_{j}\right|>T_{h}\\ 0,\;\;\;\;\;\;\;if\;\left|C_{j}\right|\leq T_{h} \end{array}$$

An improved method was proposed using Stein’s unbiased risk estimate [28]. An adaptive wavelet energy entropy method was used to filter noise in [29]. In this study, the threshed $T_{h}{}_{new}$ used a permutation entropy method based on Stein’s unbiased risk estimate [28].
(5)$$H_{pe}\left(m\right)=-\overset{m!}{\underset{j=1}{\sum }}p_{j}\ln p_{j}$$
where *m* is the dimension of time series signals and $p_{j}$ is the possibility of each permutation.

The total energy of permutation entropy is
(6)$$PE_{total}=-\overset{K}{\underset{k=1}{\sum }}\overset{m!}{\underset{j=1}{\sum }}p_{j}\ln p_{j}=\overset{K}{\underset{k=1}{\sum }}H_{pe}\left(m\right)$$

The relative permutation entropy energy is
(7)$$r_{pe}=\frac{H_{pe}\left(m\right)}{PE_{total}}$$

The new adaptive threshold $T_{h}{}_{new}$ was proposed as
(8)$$T_{h}{}_{new}=\left(\log r_{pe}\sqrt{\frac{\log\left(n\log\left(n\right)\right)}{2}}-a\right)\times b$$
where *n* is the window length of an EEG segment and *a* and *b* are two constants (9 and 6, respectively) that are calculated offline empirically. In this study, a sliding window technique with a step of one second (s) and a fixed length (*L* = 10 s) was used to read and process the raw EEG signals. Thus, there are 9 s signals that are overlapped between two sequential sliding windows.

### 2.3. Features’ Extraction

Five features were selected based on the pre-processed data to evaluate their differences in different anaesthetic states. The features are the sample entropy (SE), fuzzy entropy (FE), permutation entropy (PE), Hurst exponent (HE), and power spectral density (PSD), and their calculation methods are provided below.

Entropy is the parameter to measure the degree of irregularity of a time series signal. In this study, SE, FE, and PE are calculated to produce a new depth of anaesthesia index [17,30].
(9)$$SE=-\ln\frac{A^{m}\left(r\right)}{B^{m}\left(r\right)}$$
where $A^{m}$ is the number of template pairs having $d\left[X_{m+1}\left(i\right),X_{m+1}\left(j\right)\right]<r$ and $B^{m}$ is the number of template pairs having $d\left[X_{m}\left(i\right),X_{m}\left(j\right)\right]<r$. Tolerance *r* was chosen as 0.2 × *std* (std is the standard deviation of the time series signals), and the dimension m was chosen as
(10)$$FE\left(X,m,n,r,N\right)=\ln\emptyset^{m}\left(n,r\right)-\ln\emptyset^{m+1}\left(n,r\right)$$
where *N* is the length of the time series. Empirically, tolerance *r* was chosen as 0.15 × *std*, *n* was selected as 2, and dimension *m* was chosen as 2 to produce the best performance.
(11)$$PE=\frac{\sum_{j=1}^{J}P_{j}\ln P_{j}}{\ln\left(J\right)}$$
where embedding dimension *m* and time delay τ are 4 and 1, respectively.

The Hurst exponent, a numerical method, was used to predict the trend in EEG signals. The time series signals *Y* with the length of *N* were divided into components $y_{m}=\left\{y_{1,n},y_{2,n},\ldots ,y_{m,n},\right\}$ and each component has the same length $m\epsilon \left\{N,\frac{N}{2},\frac{N}{4},\ldots \right\}$. The following procedure was used to calculate the range response in the Hurst method.

Calculating the mean value of each component:


(12)
$$mean=\frac{1}{m}\overset{m}{\underset{i=1}{\sum }}y_{i,n}$$


2.Creating a mean-adjusted series:


(13)
$$X_{i,n}=y_{i,n}-mean,\;\;\;\;\;\;for\;i=1,2,\ldots ,m$$


3.Calculating the cumulative deviate series:


(14)
$$Z_{i,n}=\overset{i}{\underset{t=1}{\sum }}X_{i,n}$$


4.Computing the ranges *R:*


(15)
$$R_{n}=\max\left\{Z_{1,n},Z_{2,n},\ldots ,Z_{m,n}\right\}-\min\left\{Z_{1,n},Z_{2,n},\ldots ,Z_{m,n}\right\}$$


5.Computing the standard deviation $S_{n}$:
(16)$$S_{n}=\sqrt{\frac{1}{m}}\overset{m}{\underset{i=1}{\sum }}{\left(y_{i,n}-mean\right)}^{2}$$

6.Calculating the rescaled range:


(17)
$${\left(R/S\right)}_{n}=\frac{1}{\left(\frac{N}{m}\right)}\overset{\left[\frac{N}{m}\right]}{\underset{n=1}{\sum }}(R_{n}/S_{n})$$


7.Calculating the rescaled range:


(18)
$$R_{r}=\underset{n=1,2,\ldots }{\min}\{mean(R_{n})\}$$


Multiple signal classification (MUSIC), as the method of an eigenvector-based frequency estimator, is used. The feature extraction was from the power spectral density (PSD) of wavelet coefficients, and the method is described below based on [14,31]:

1.$A_{j}$ and $D_{j}$ were used as input signals for the eigenvector method:

(19)$$SD:\{\begin{array}{c} D_{j}\to P\left(D_{j}\right)=ED_{j}\\ A_{j}\to P\left(A_{j}\right)=EA_{j} \end{array}$$
where j was set as six-level wavelet decomposition, the order of 16 Daubechies wavelet filter was used [32], and the principal eigenvector was 6.

2.Means of $EA_{j}$ and $ED_{j}$ are as follows:



(20)
$$Mean:\{\begin{array}{c} D_{j}\to M(ED_{j})=mean\;(ED_{j})\\ A_{j}\to M(AD_{j})=mean\;(AD_{j}) \end{array}$$



3.Standard deviation (*STD*) of $EA_{j}$ and $ED_{j}$ are as follows:



(21)
$$STD:\{\begin{array}{c} D_{j}\to S(ED_{j})=std(ED_{j})\\ A_{j}\to S(AD_{j})=std\;(AD_{j}) \end{array}$$



4.Derived from steps 2 and 3:



(22)
$$D:\{\begin{array}{c} \;\;M_{j}\to \frac{1}{2}\{\log\left(mean\left(M(ED_{j}\right)\right))+\\ log\left(mean\left(M(AD_{j})\right)\right)\}\\ S_{j}\to \frac{1}{2}\{\log\left(mean\left(S(ED_{j}\right)\right))+\\ log\left(mean\left(S(AD_{j})\right)\right)\} \end{array}$$



5.Deriving the feature:

(23)$$Fea=\frac{\;k_{1}\times M_{j}+k_{2}\times \;S_{j}}{k_{3}}$$
The constant parameters $k_{1}$, $k_{2}$, and $k_{3}$ were chosen as 28, 90, and 3, respectively.

These five features were extracted individually after pre-processing the raw EEG signals. Then, they were used as the input data to train a robust model to classify different states of anaesthesia.

### 2.4. Regression Models and Evaluation Measures

In this study, a Gaussian process regression model, as a nonparametric Bayesian approach, was applied. This kind of regression model has its advantage on small datasets and provides reliable uncertain predictions. We also utilized linear regression, SVM, and regression tree models for comparison and evaluation. The dataset with 73 subjects was divided into the training and testing sets. A total of 60 subjects were randomly selected for training, and the aforementioned five features of each subject were calculated for training and evaluating regression models to find the best performance models based on different combinations of features.

The training models were evaluated using R-square ($R^{2})$, root-mean-square-error (RMSE), mean-square-error (MSE), and mean absolute error (MAE).
(24)$$R^{2}=1-\frac{\sum_{i}{\left(y_{i}-f_{i}\right)}^{2}}{\sum_{i}{\left(y_{i}-\bar{y}\right)}^{2}}$$
where $y_{i}$ is the data set value (BIS value), $\bar{y}$ is the mean of the BIS values, and $f_{i}$ is the calculated value from regression results. The range of the $R^{2}$ values are between 0 and 1. The higher value of $R^{2}$ indicates a higher correlation between the BIS value and the extracted feature values, and vice versa.
(25)$$MSE=\frac{1}{n}\overset{n}{\underset{i=1}{\sum }}{\left(Y_{i}-\bar{Y_{i}}\right)}^{2}$$
(26)$$MAE=\;\frac{\sum_{i=1}^{n}\left|Y_{i}-\bar{Y_{i}}\right|}{n}$$
where *n* is the number of the data points, $Y_{i}$ is a set of the observed values, and $\bar{Y_{i}}$ is a set of the regression results.

After the best performance model was selected and evaluated using the testing datasets, the Bland–Altman method was used to assess the degree of agreement between the proposed index, $N_{DoA}$, and the BIS index. In addition, the Pearson correlation coefficient was also applied to assess the correlation of $N_{DoA}$ with the BIS index and clinical observations.
(27)$$r=\frac{\sum^{\;}\left(x_{i}-\bar{x}\right)\left(y_{i}-\bar{y}\right)}{\sqrt{\sum^{\;}{\left(x_{i}-\bar{x}\right)}^{2}{\left(y_{i}-\bar{y}\right)}^{2}}}$$

### 2.5. Real-Time DoA Monitor under Sliding Window Framework

In order to have a real-time response, the sliding window technique was applied and the window size was chosen as 10 s, and the overlap between two adjacent windows was 9 s. In addition, the tuned index, $DoA_{tuned}$, was calculated using Equation (27), from the mean value between previous and present data.
(28)$$DoA_{tuned}=0.8\times Index_{pre}+0.2\times Index_{cur}$$
where $Index_{pre}$ is the mean value of the DoA index from the last four seconds, $Index_{cur}$ is the current DoA index, and $DoA_{tuned}$ presents the real-time index second by second. Using this sliding window method, the DoA index could be updated every second. Consequently, the time delay for monitoring the DoA could be reduced. Meanwhile, the starting time of processing the recording raw EEG data was from the fifth second, and there was a four-second delay.

## 3. Results

### 3.1. Raw EEG Signal Pre-Processing and Features’ Selection

Figure 2 presents the sample denoising results with the proposed threshold $T_{hnew}$ using the permutation entropy method. The raw EEG signal (from patient ID: L112161431) has low amplitude noise (between 2000 s and 2010 s) and spike noise (between 2980 s and 2990 s), as shown in Figure 2a–c, respectively. Compared with the denoised EEG signal using the threshold $T_{h}$, as shown in Figure 2d,e, the denoised EEG signal using the proposed threshold $T_{hnew}$, as shown in Figure 2f,g, has much less noise. After using both denoising threshold methods, the EEG signal has a similar amplitude. However, the proposed threshold method in this study removed almost all the noise, as shown in Figure 2d–g.

There are EEG signals from two channels (CH1 and CH2) in the datasets. In this study, CH2 data were selected because their correlation between the features and anaesthetic states was higher than that of CH1. The five features of fuzzy entropy, permutation entropy, sample entropy, power spectral, and Hurst range response were extracted from the denoised CH2 EEG signal. All of the features were measured using coefficients of determination ($R^{2}$), which reflects how close a feature is to different anaesthetic states. The range of $R^{2}$ values is between 0 and 1. A higher value of $R^{2}$ indicates a higher correlation between the BIS and the extracted feature, and vice versa. As an example of the patient ID: L01040838, the values of $R^{2}$ are 0.7497, 0.6974, 0.7034, and 0.6214 for fuzzy entropy, permutation entropy, sample entropy, and power spectral, respectively. The feature of Hurst range response has a relatively lower correlation with the BIS compared with the other four features. However, the Hurst range response has an obvious state change point from consciousness to unconsciousness at 762 s, as shown in Figure 3.

### 3.2. Regression Model Training

A total of 60 subjects were randomly selected as the training set from 73 subjects. The total EEG recording time for the 60 subjects was 385,116 s, with the corresponding BIS values for each second. The remaining 13 subjects were used to evaluate the training regression models. Fivefold cross-validation, as a resampling procedure, was used to protect against overfitting by partitioning the data sets.

The MATLAB regression learner app [32] was used for training with four different models, including linear regression, trees, SVM, and Gaussian process models. Table 2 lists the best performance for each type of training model. For example, fine Gaussian SVM has the greatest value of $R^{2}$ and smallest value of RMSE among linear SVM, quadratic SVM, cubic SVM, medium SVM, and coarse SVM. The results in Table 2 show that the squared exponential Gaussian process regression model (SEGP-RM) has a better overall performance. Thus, the SEGP-RM was selected for training the robust model. Figure 4 and Figure 5 show the fivefold cross validation results of the SEGP-RM. As a result, $R^{2}$ is 0.76, while RMSE, MSE, and MAE are 6.18, 38.22, and 5.15, respectively. The response plot is used to explore the correlation between the predicted values and true values, and the correlation coefficient is 0.89. The predicted plot versus actual plot in Figure 4 shows how well the regression model makes predictions for different response values. A perfect regression model has a predicted response equal to the true response, so all of the points (blue points in the below figure) lie on a diagonal line (red line in the below figure). The vertical distance from the line to any point is the prediction error for that point.

### 3.3. Performance of the Training Models and Results of the Nex Index

To further validate the performance of the trained models, the remaining 13 subjects were used for evaluation. The metric, Pearson correlation coefficient, which is defined in Equation (26), is used to evaluate the four trained regression models listed in Table 2.

Figure 6 indicates the Pearson correlation coefficients of the 13 testing subjects. Patient number 14 represents the average value of all testing data results. The correlation coefficient of the SEGP-RM is higher than the other three regression models, robust linear, fine trees, and fine Gaussian SVM. The highest correlation coefficient is 0.940 for patient No. 8 (subject ID: L01131248), and the lowest one is 0.580 for patient No. 3 (subject ID: L01060841). Meanwhile, the average value of correlation coefficients from these 13 testing subjects is 0.812.

The difference between $N_{DoA}$ and BIS is defined as $d=N_{DoA}-BIS$. The mean value of the difference between $N_{DoA}$ and BIS is diff=mean(d) and the standard deviation of the difference is sd=std(d). The Bland–Altman method suggests that 95% of the d should lie between diff+2sd and diff−2sd (two black lines in Figure 7a) if the data d have a normal distribution. In Figure 7b, the pattern matches well with the distribution of d. The Bland–Altman plot of the data from 13 testing subjects, calculates the mean difference diff=2.1348 as the ‘bias’ (red dash line). For the 95% agreement limitation, the upper limitation is diff+2sd=21.46 and the lower one is diff−2sd=−17.19. The agreement rate between $N_{DoA}$ and BIS is 94.91%, as shown in the scatter plot in Figure 7a.

## 4. Results and Discussion

### 4.1. Length of Sliding Window

During surgery and clinical operations, the whole anaesthetic process is controlled by anaesthetists. The surgery’s incisions started when the patient’s anaesthetic states were considered at the middle anaesthesia stage (BIS index between 40 and 60). The real-time technique proposed in this paper only has four seconds’ delay at the beginning of recording data (awake stage), which may not affect clinical data processing and anaesthetic state tracing in clinical applications. The sliding window size affects the real-time results of the anaesthesia monitoring and efficiency. If the sliding window size is too big, the results may cause a delay in the clinical operations, and too small of a window size may affect the feature extractions during the transitions of two anaesthesia states. In this study, a fixed-length sliding window of 10 s was selected.

### 4.2. The Effect of Signal Quality

The signal quality indicator is an index recorded with BIS values in real time, which measures the quality of the raw EEG signals. Artifacts, such as eye movement, muscle activity, head motion, and the electrical knife operations, affect the impedance calculation and result in poor signal quality. We found that the BIS values were not displayed on the monitor screen when SQI was lower than 15. Meanwhile, we reviewed the raw data log files, and the corresponding BIS values are −3276.8 when the signal quality is poor (SQI lower than 15). However, our proposed real-time monitor index, $N_{DoA}$, can still provide the corresponding anaesthetic states in the case of poor signal quality.

Figure 8 demonstrates a sample case with poor signal quality (subject ID: L01121339). The BIS values are −3276.8 during the time from 369 to 371 s, from 1708 to 1710 s, and from 1790 to 2017 s. However, the proposed index, $N_{DoA}$, computed and provided the DoA values clearly during the period. Figure 9 presents another case with poor signal quality (subject ID: L01210841). The subject was a 17-year-old male patient weighing 46 kg. The intraoperative EEG monitoring in surgery started at 08:41:41 and finished at 13:37:27. The detail of anaesthesia medications for this patient is reported in Table 3. In this case, the SQI index was below 55 in the awake stage (from 1 to 610 s). Furthermore, the invalid BIS index (−3276.8) could not display on the screen monitor as poor signal quality (SQI < 15) during these periods from 89 s to 132 s and 389 s to 412 s, while the proposed $N_{DoA}$ was able to estimate the DoA. Compared with the BIS index, the proposed method has a better correlation and agreement with clinical observations during periods of poor signal quality. Moreover, two additional cases with the same phenomenon could be observed in Figure 10 and Figure 11.

### 4.3. The Effect of EMG Signals

EMG signals, caused by muscle activities, were recorded as a spark from forehead muscles areas. These signals can have a negative effect on increasing the value of the BIS index, and the neuromuscular blocking agents can reduce or eliminate this side effect [33]. In Figure 12 there are two spikes with the BIS index higher than 70 in a general anaesthesia state from 2560 s to 2580 s and from 2870 s to 2890 s. The index of EMG signals also reflected this fact. Meanwhile, the SQIs are between 90 and 100 during these two periods, which indicates reliable and accurate signal quality with the BIS index. Therefore, the EMG activities during these periods might result in two spikes for the BIS indexing. However, as observed from the waveform of the $N_{DoA}$, the trend of the proposed index does not change in the general anaesthesia state.

## 5. Conclusions

This paper studies the real-time monitoring of anaesthesia states using hybrid statistical features and machine learning methods. Firstly, a wavelet threshold method was used to remove the low-amplitude and spike noise interference. Then, hybrid statistical features were extracted from the denoised EEG signals. Next, the regression models were trained using these extracted features by applying the machine learning methods, linear regression, SVM, regression trees, and Gaussian process regression. The squared exponential Gaussian process regression model outperformed the other models. This yields a better result in accuracy and a higher degree of agreement rates. In addition, the proposed index, $N_{DoA}$, has a higher correlation with anaesthetists’ observations than the BIS index during periods of poor signal quality.

The results were evaluated by comparing the correlation and agreement with the BIS index (current gold standard) and the clinical observations recorded by the attending anaesthetists. The results of $N_{DoA}$ are highly correlated with the BIS index. The highest correlation coefficient, $R^{2},$ is 0.940, and the average is 0.812. Furthermore, the agreement measured by the Bland–Altman method presented a higher degree of agreement between the proposed $N_{DoA}$ and the BIS index. The scatter plot shows an agreement of 94.91%.

## Figures and Tables

**Figure 1 sensors-22-06099-f001:**
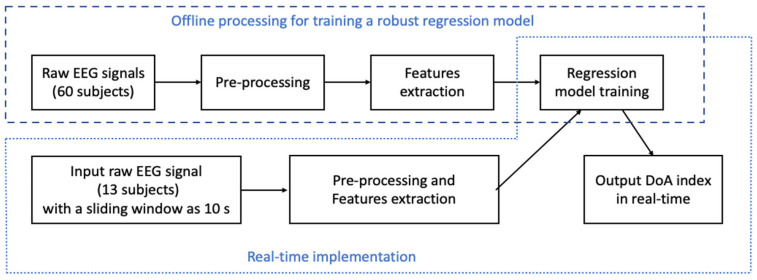
The diagram of the new DoA index development.

**Figure 2 sensors-22-06099-f002:**
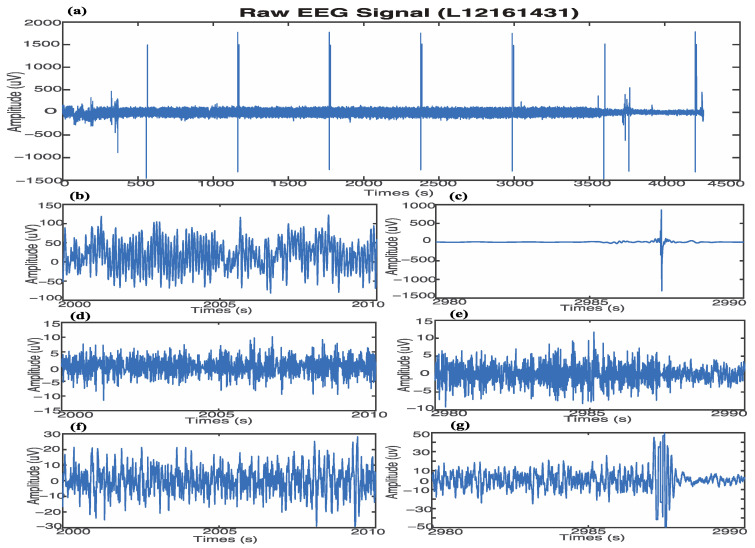
Results of denoising raw EEG data, including the low amplitude (time is between the 2000 s and 2010 s) and spike noise (time is between 2980 s and 2990 s). (**a**) Raw EEG data, patient ID: L12161431. (**b**) Raw EEG data with low amplitude noise. (**c**) Raw EEG data having spike noise. (**d**) EEG signal after denoising low amplitude with threshold $T_{h}$. (**e**) EEG signal after denoising spike with threshold $T_{h}$. (**f**) EEG signal after denoising low amplitude with proposed threshold $T_{hnew}$. (**g**) EEG signal after denoising spike with proposed threshold $T_{hnew}$.

**Figure 3 sensors-22-06099-f003:**
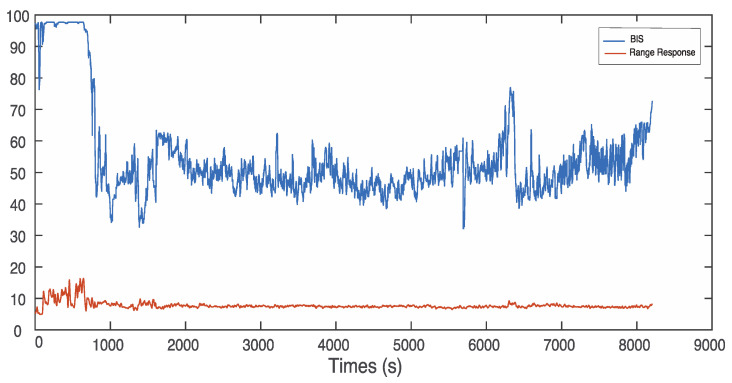
Relationship between the Hurst range response value and the BIS value.

**Figure 4 sensors-22-06099-f004:**
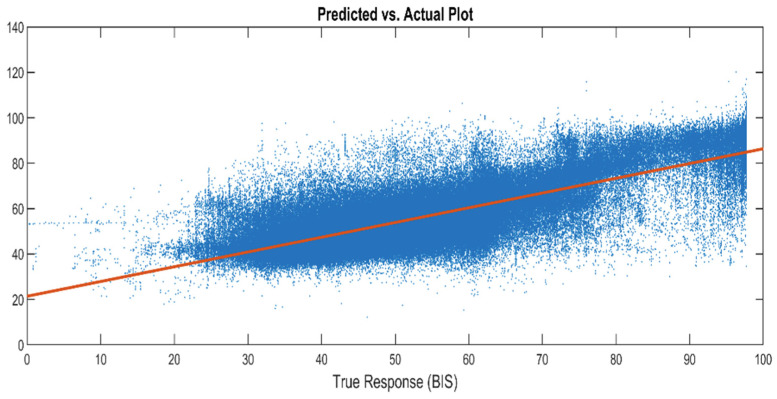
Predicted response from the SEGP-RM vs. actual values (BIS).

**Figure 5 sensors-22-06099-f005:**
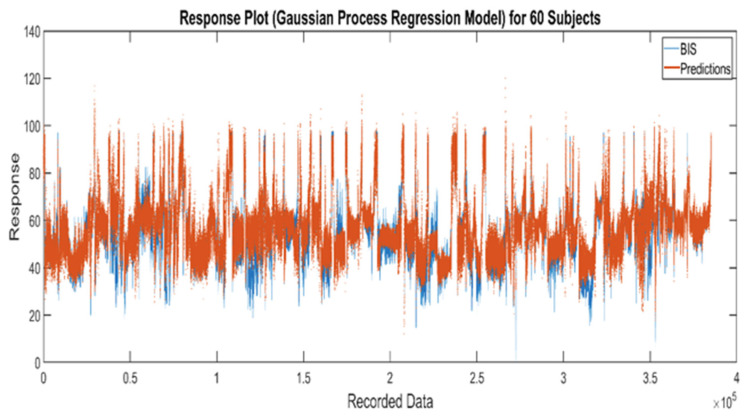
Response plot of the SEGR-RM for the 60 training subjects.

**Figure 6 sensors-22-06099-f006:**
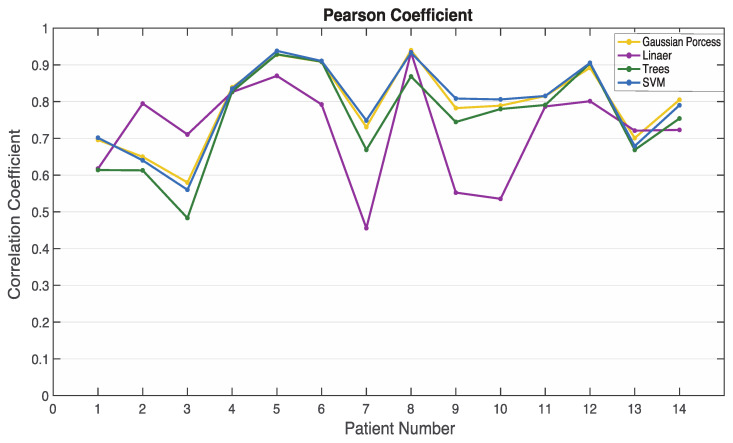
Pearson correlation coefficients of the testing results and BIS values for 13 subjects (patient 14 represents the average results of the 13 testing subjects).

**Figure 7 sensors-22-06099-f007:**
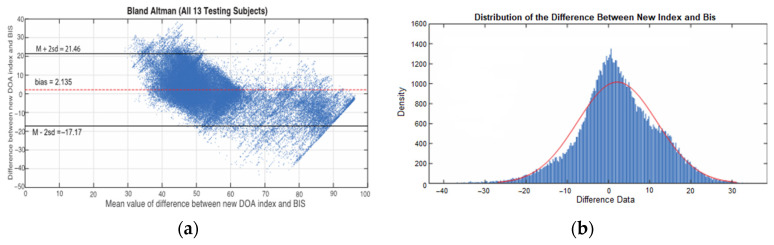
(**a**) Bland-Altman plot between $N_{DoA}$ index and BIS. (**b**) Distribution plot between new DoA index and BIS.

**Figure 8 sensors-22-06099-f008:**
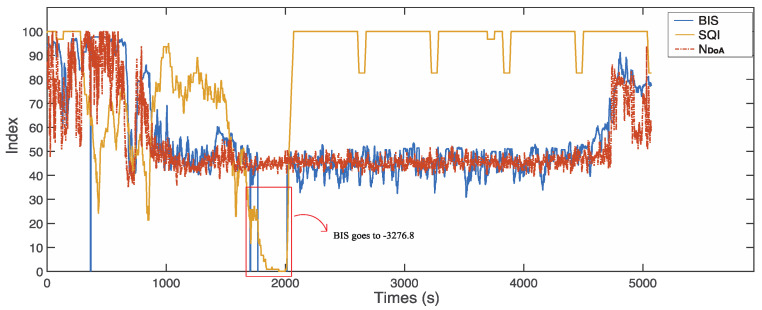
The proposed index, $N_{DoA}$, can monitor the anaesthetic states change, but BIS cannot when SQI is lower than 15 (subject ID: L01121339).

**Figure 9 sensors-22-06099-f009:**
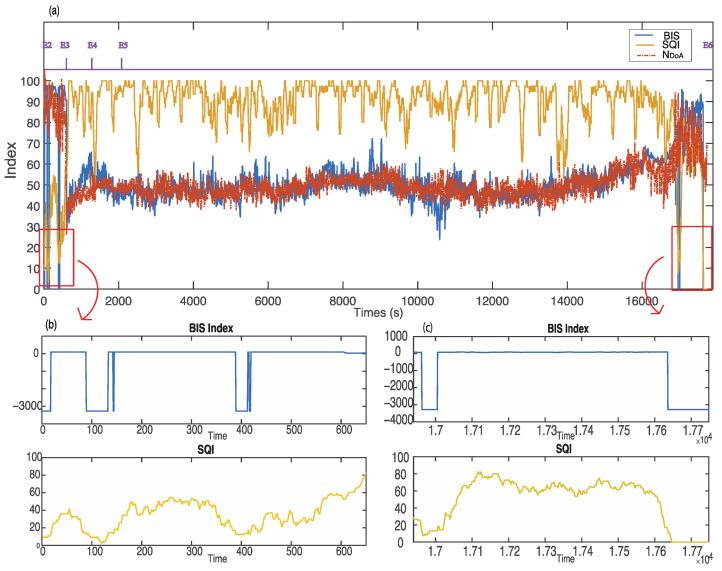
(**a**) Comparison between $N_{DoA}$ and BIS in the case of poor signal quality (subject ID: L01210841). (**b**) Zoom out during the period between 0 and 650 s. (**c**) Zoom out during the period between 16,940 and 17,748 s.

**Figure 10 sensors-22-06099-f010:**
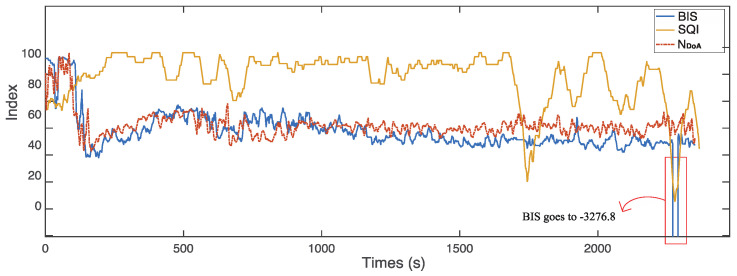
Comparison between $N_{DoA}$ and BIS in the case of poor signal quality (subject ID: L01131131).

**Figure 11 sensors-22-06099-f011:**
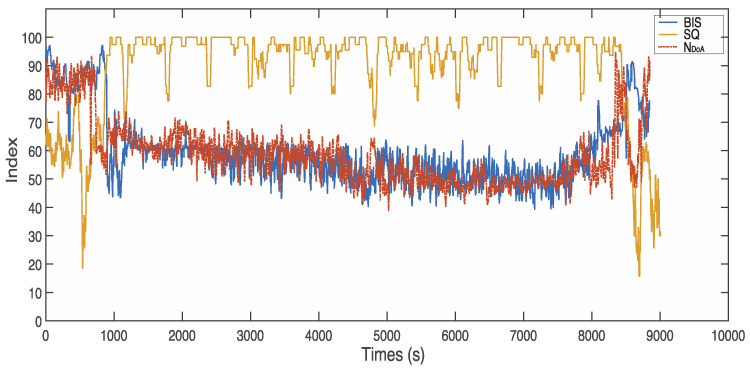
Comparison between $N_{DoA}$ and BIS in the case of poor signal quality (subject ID: L01131400).

**Figure 12 sensors-22-06099-f012:**
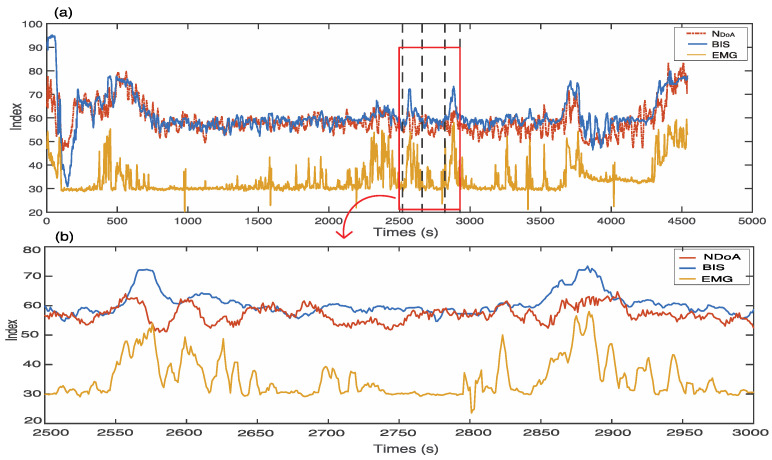
(**a**) EMG causes the BIS index to increase at two spikes, and the trend of $N_{DoA}$ does not change during these periods. Subject ID: L01041002. (**b**) Zoom out during the period between 2500 and 3000 s.

**Table 1 sensors-22-06099-t001:** Summary of patient demographics.

Age (year)	Weight (kg)	Height (cm)	Gender (M/F)
2–83	11–101	90–194	39/34

**Table 2 sensors-22-06099-t002:** Training model regression of the anaesthetic states.

Training Model	$R^{2}$	RMSE	MSE	MAE
Robust Linear	0.43	11.21	125.58	8.81
Fine Tress	0.6	9.41	88.53	6.25
Fine Gaussian SVM	0.73	6.61	43.67	5.12
Squared Exponential Gaussian Process Regression	0.79	5.66	32.04	4.84

**Table 3 sensors-22-06099-t003:** Details of anaesthesia medication.

Subject	Event	Timestamp	Time (s)
L01210841	E1. Add Dexmedetomidine Hydrochloride (30 ug)	08:25:00	-
	E2. Intraoperative EEG monitoring	08:41:41	0
	E3. Add Dezocine (6 mg)	08:55:00	799
	E4. Add Tropisetron Hydrochloride (2 mg)	09:05:00	1399
	E5. Add Ketorolac (30 mg)	09:16:00	2059
	E6. Surgery End	13:37:27	17,746

## Data Availability

The data and materials used in this study are available at the University of Southern Queensland under the research data management policy.

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
