# Peer review of "Real-Time Depth of Anaesthesia Assessment Based on Hybrid Statistical Features of EEG"

_sensors, 2022, doi:10.3390/s22166099_

Round 1
Reviewer 1 Report
The manuscript describes a new DoA method for monitoring the anaesthesia based on EEG signals. The noise in the EEG signals were removed by the wavelet transform method. The entropy features were extracted on the noise-free EEG signals. The squared exponential Gaussian process regression model was used to identify the states of anaesthesia. The experimental results were validated with the correlation coefficient and Bland-Altman agreement evaluation methods. In general, it is an interesting application for anaesthesia monitoring. The following comments could be considered for manuscript improvement:
(1) In Abstract, it is normally variables would not be recommended to appear in Abstract section. The abstracting index system and web page are difficult to correctly display the symbols and variables. It is suggested replacing the variables with text descriptions. The text of "Five eigenvalues were used to extract DoA features" is better to revised, because the entropy features did not compute the "egienvalues.
(2) Informed consent statement: the manuscript declares not applicable. In Section 2.1, the authors declares the EEG and BIS datasets of 73 subjects were collected from several hospitals. It is better to provide the informed consent, or declare why is not applicable. In addition, the basic demographic information such as gender and age statistics should be provided.
(3) In Figure 1, the "real-time monitoring the states of anaesthesia" is given. According to the signal preprocessing based on wavelet transform and the extraction of entropy features, these methods are offline signal processing and analysis methods, and cannot support the real-time applications.
(4) In Equation 27, the coefficients of the mean values of DoA index from the last four seconds and the current DoA are 0.8 and 0.2, respectively. The question is there any criterion to determine these coefficient, or if they are optimal for the present application?
(5) There were 60 subjects randomly selected from the total 73 subjects for training purpose, and the resting 13 subjects were used for testing evaluation. The ratio of testing data is relatively low, and it is suggested testing vs. training ratio to be over 20%.
(6) Regarding Table 1 and Figure 5, how about the testing results?
Author Response
The response letter to reviewer 1 is in the attachment.

Reviewer 2 Report
The paper is devoted to EEG signal processing to identify the depth of anesthesia. The authors proposed a new DoA index using a wavelet transform of EEG signal and extracting DoA features using five different algorithms.
The authors used a sliding window technique for the proposed threshold wavelet denoising method. However, it is not clear from the text what is the reasoning for setting the sliding window parameters, viz., 10 s widow size and 9s overlap. What is the influence of these parameters on denoising method efficiency? Have other values been examined?
I’m afraid 73 subjects used for method verification in this study is a relatively small dataset for medicine. I suggest using a larger dataset to evaluate efficiency metrics and confidence interval more reliably in future work.
How and with what assumptions and limitations were the constants a and b from Eq. (8) calculated?
Fig. 2 (c) and (e) have substantially different scales, so it’s hard to compare the denoised signal with the raw one.
Line 230: “the red line is the linear correlation with the BIS” – which red line?
Figure 4 is hard to analyze: e.g., if the predicted values should match the actual response, one expects to see that, e.g., for the true response value of 60, we obtain a predicted value close to 60. However, from Fig. 4, I can conclude that the predicted values are almost equally distributed between 30 and 90. What does it mean? The red line has a slope of less than 45 deg. as expected, and, e.g., an actual value of 90 in the red line in Fig. 4 indicates the predicted value to be about 80. Please add some discussion to the text concerning this figure.
Besides, Fig. 6 shows a substantial spread in the Pearson correlation coefficient for different patients, varying from as low a value as 0.58 to almost 0.94. So, I doubt the reproducibility and reliability of the proposed DoA index for practical use. Otherwise, it should be indicated and discussed in the text of the paper. I’m afraid the average value, as shown in line 268, is not the best case for medical tasks. Again, I suggest using a larger dataset to verify the proposed approach reliably.
The comparison of the proposed NDOA index with the known BIS index approach is not highlighted, and the practical advantages of using the developed method are unclear. Bland-Altman method and the Pearson correlation coefficient show that the proposed index has almost the same value as the BIS index. So, what is the advantage of using the proposed approach?
Some minor English improvements and spellchecking are needed, e.g.:
Line 104: replace the comma with a full stop or lowercase for the word “Then”.
Line 155: missing predicate.
Author Response
The response letter to reviewer 2 is in the attachment.

Round 2
Reviewer 1 Report
The revision manuscript has greatly improved with the corresponding responses to the review comments.
The additional revision of Figure 1 should be done: The block diagram of "Output DoA index in real-time" should be correctly displayed without the editable mode (see Figure 1, Page 2).
Author Response
Thanks for your reminder.
We have corrected figure 1 in the manuscript.